 **eLIFE**

# Stimulus-dependent recruitment of lateral inhibition underlies retinal direction selectivity

**Qiang Chen, Zhe Pei[†], David Koren, Wei Wei\***

Department of Neurobiology, The University of Chicago, Chicago, United States

**Abstract** The dendrites of starburst amacrine cells (SACs) in the mammalian retina are preferentially activated by motion in the centrifugal direction, a property that is important for generating direction selectivity in direction selective ganglion cells (DSGCs). A candidate mechanism underlying the centrifugal direction selectivity of SAC dendrites is synaptic inhibition onto SACs. Here we disrupted this inhibition by perturbing distinct sets of GABAergic inputs onto SACs – removing either GABA release or GABA receptors from SACs. We found that lateral inhibition onto Off SACs from non-SAC amacrine cells is required for optimal direction selectivity of the Off pathway. In contrast, lateral inhibition onto On SACs is not necessary for direction selectivity of the On pathway when the moving object is on a homogenous background, but is required when the background is noisy. These results demonstrate that distinct sets of inhibitory mechanisms are recruited to generate direction selectivity under different visual conditions.

**\*For correspondence:** weiw@uchicago.edu

**Present address:** [†]Sophie Davis School of Biomedical Education, The City College of New York, NewYork, United States

**Competing interests:** The authors declare that no competing interests exist.

## Introduction

Encoding of motion direction first appears in the inner plexiform layer (IPL) of mammalian retina (*Barlow and Hill, 1963*; *Barlow and Levick, 1965*; *Oyster and Barlow, 1967*), where positive- and negative-contrast motion stimuli are processed in anatomically distinct On and Off sublaminae. One of the main output neurons that signal motion direction from the retina to higher visual centers, the On-Off DSGC, has bistratified dendritic arbors that receive directionally tuned inhibition from On and Off subtypes of SACs at each of these sublamina (*Figure 1A*) (*Famiglietti, 1983, 1992*; *Kittila and Massey, 1997*; *Taylor and Vaney, 2002*). SACs are monostratified, axonless inter-neurons that release GABA and acetylcholine from varicosities at the distal ends of their dendritic arbors (*Brecha et al., 1988*; *Famiglietti, 1991*; *Kosaka et al., 1988*; *O'Malley and Masland, 1989*; *Vaney and Young, 1988*). Generating directionally tuned GABAergic inhibition onto DSGCs requires two mechanisms. First, GABAergic inputs preferentially originate from SAC dendritic sectors that extend in the anti-preferred (null) direction of On-Off DSGCs (*Figure 1B*, *Briggman et al., 2011*; *Fried et al., 2002*; *Lee et al., 2010*; *Wei et al., 2011*). Second, the dendritic sectors of SACs are electrotonically isolated and directionally tuned to motion in the centrifugal direction (from soma to dendritic tips) (*Figure 1B*, *Euler et al., 2002*). Since null-direction motion for a DSGC corresponds to centrifugal motion for its presynaptic SAC dendrites, maximal GABA release from SACs to DSGCs occurs during motion in the null direction (*Figure 1B*).

Along with providing inhibition to DSGCs, SACs at each sublamina receive GABAergic lateral inhibition from neighboring SACs (*Ding et al., 2016*; *Kostadinov and Sanes, 2015*; *Lee and Zhou, 2006*) and other wide-field amacrine cells (*Ding et al., 2016*; *Lee and Zhou, 2006*). The role of this lateral inhibition in establishing the centrifugal preference of SACs remains controversial (*Ding et al., 2016*; *Euler et al., 2002*; *Hausselt et al., 2007*; *Lee and Zhou, 2006*; *Münch and Werblin, 2006*; *Oesch and Taylor, 2010*). Centrifugal direction selectivity has also

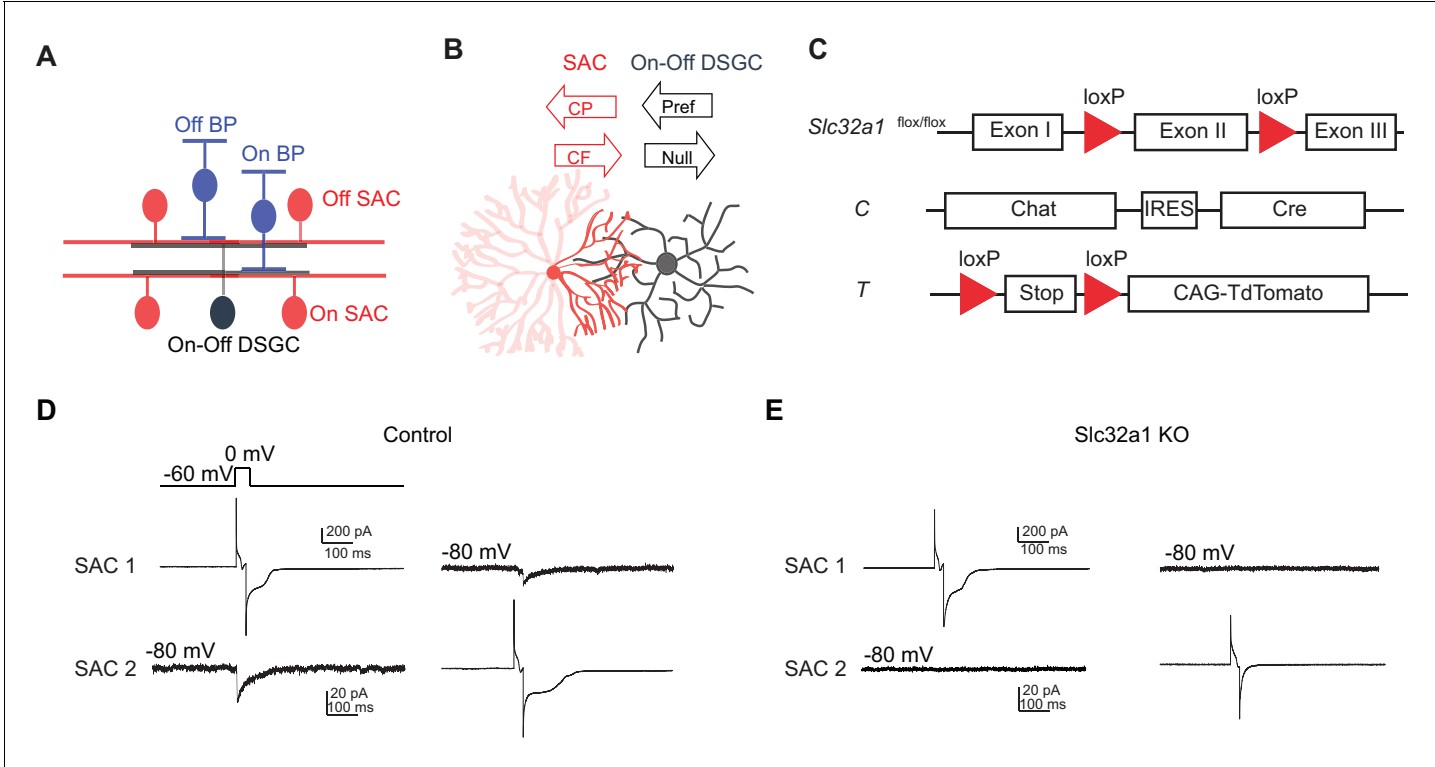

**Figure 1.** Genetic deletion of *Slc32a1* from SACs eliminates reciprocal inhibition between SACs. (**A**) Schematic shows laminar organization of cell types involved in the direction selective circuit in the IPL. (**B**) Schematic shows the orientation of SAC and DSGC dendrites connected by GABAergic synapses. Black arrows indicate referred and null directions of DSGC. GABAergic inputs onto DSGC come from SAC dendritic quadrant (dark red) that extends to the null direction of the DSGC. Red arrows indicate centrifugal (CF) and centripetal (CP) direction of motion for the SAC quadrant. Maximal activation and GABA release occurs in the CF direction, while minimal activation occurs in the CP direction. (**C**) Schematic diagram shows transgenes involved in *Slc32a1* KO mice. (**D**) Voltage clamp traces from reciprocal On SAC-SAC pairs in control mouse (*CT*) show IPSCs evoked in one SAC at −80 mV by depolarizing the other in the presence of glutamatergic and nicotinic receptor antagonists (see Experimental procedures). (**E**) As in D but for *Slc32a1* KO mouse (*Slc32a1* ^flox/flox^ *CT*).

been attributed to other mechanisms such as patterned distribution of voltage-gated channels (*Euler et al., 2002*; *Hausselt et al., 2007*; *Oesch and Taylor, 2010*) and chloride transporters (*Gavrikov et al., 2006*), segregation of excitatory inputs from distinct bipolar cell types (*Kim et al., 2014*); but see *Stincic et al., 2016*), and skewed distribution of glutamatergic inputs along SAC dendrites (*Vlasits et al., 2016*). However, the relative contributions of these synaptic and intrinsic mechanisms to SAC centrifugal preference is unclear. Furthermore, the role that synaptic inhibition onto SACs plays in the direction selectivity of DSGCs has not yet been experimentally demonstrated.

To address these outstanding questions, we perturbed GABAergic inhibition onto SACs using two types of SAC-specific genetic manipulations. The first one disrupted GABA release from SACs by conditionally knocking out the vesicular GABA transporter (Vgat) gene *Slc32a1* in SACs. This selectively removed reciprocal SAC-SAC inhibition and allowed us to determine its contribution to the centrifugal direction selectivity of SACs. The second manipulation blocked all GABAergic inputs onto SACs by removing their functional GABA receptors. This enabled us to determine the role that total GABAergic inhibition onto SACs plays in the direction selectivity of both SACs and On-Off DSGCs. Combining these genetic manipulations with pharmacology, we found that motion stimuli with different contrast and backgrounds recruit different sets of inhibitory mechanisms for the computation of motion direction.

## Results

### Direction selectivity of SAC dendrites persists in the absence of SAC-mediated GABA release during a simple moving bar stimulus

To eliminate reciprocal GABAergic inhibition among neighboring SACs, we used a conditional knockout (KO) mouse line in which *Slc32a1* is selectively removed from SACs (*Slc32a1* KO) to block GABA release from SACs. This line contains homozygous floxed *Slc32a1* alleles (acronym: *Slc32a1-flox/flox*) to replace the endogenous *Slc32a1* gene, and a choline acetyltransferase (*Chat*)-*IRES-Cre* knockin allele for SAC-specific *Cre* expression (acronym: *C*). A floxed *tdTomato* transgene (acronym: *T*) was also included when SACs were targeted for electrophysiological recordings (*Figure 1C*). We have previously shown that these KO mice display disrupted GABAergic inhibition from SACs to DSGCs and no detectable developmental compensation (*Pei et al., 2015*). To confirm that GABAergic synapses between neighboring SACs are also lost in these mice, we performed paired recordings from neighboring SACs in the ganglion cell layer with intersoma distance of 50~100 um. Evoked GABAergic inhibitory postsynaptic currents (IPSCs) were measured from one SAC at holding potential of −80 mV while the other SAC was depolarized to 0 mV for 20 ms in voltage clamp configuration. We detected reciprocal inhibition between SACs in 9 out of 9 pairs in the control group (*Figure 1D*, mean peak amplitude 16.4 ± 2.8 pA, five mice), consistent with previous studies (*Kostadinov and Sanes, 2015*; *Lee and Zhou, 2006*). We did not detect any evoked IPSCs in the KO group (*Figure 1E*, n = 9 pairs, three mice), indicating that reciprocal SAC-SAC inhibition is abolished in *Slc32a1* KO mice.

Next, we examined centrifugal direction selectivity of SAC dendrites in these KO mice using two-photon imaging of the calcium indicator GCaMP6. We imaged calcium transients in varicosities located in the furthest distal 20 µm of GCaMP6-expressing SAC dendrites while a bright bar moved against a homogeneous dark background (*Figure 2A*). We term this stimulus "simple moving bar stimulus". As expected, the leading edge of the bright bar evoked calcium transients in On SACs and the trailing edge evoked responses in Off SACs (*Figure 2B*, and *Videos 1* and *2*). We used the peak amplitude of the calcium transients in varicosities as a measure of the strength of dendritic activation. We quantified direction selectivity of responses using the direction selectivity index (DSI, see Experimental methods) and the vector sum. In control mice, dendritic calcium signals in both On and Off SACs show strong directional tuning to motion in the centrifugal direction (*Figure 2B and C*). In *Slc32a1* KO mice, centrifugal direction selectivity (both DSI and vector sum) of On SACs is similar to that of control mice (*Figure 2D*). However, Off SACs exhibit a small, but statistically significant, increase in DSI and vector sum values (*Figure 2F*) due to decreased centripetal direction response (*Figure 2—figure supplement 1*). Adding the GABA$_A$ receptor antagonist SR95531 to *Slc32a1* KO retinas reduces these direction selectivity parameters for both On and Off SACs (*Figure 2D–2G*), indicating a role for GABAergic inhibition of SACs by non-SAC amacrine cells. Our results demonstrate that, for a simple moving bar stimulus, losing reciprocal SAC-SAC inhibition in *Slc32a1* KO mice does not affect the direction selectivity of On SACs, and enhances the direction selectivity of Off SACs.

### SAC-specific deletion of GABA$_A$ receptors

The GABAergic inhibition required for generating the centrifugal direction selectivity of On and Off SACs could come from two sources besides other SACs. The first is the GABAergic synapse onto presynaptic bipolar cells from other amacrine cell types (*Hoggarth et al., 2015*; *Lee and Zhou, 2006*). The second is direct GABAergic input onto SACs from non-SAC amacrine cells (*Ding et al., 2016*). To distinguish between these two possibilities, we examined the direction selectivity of SACs when GABA$_A$ receptors are removed from SACs but not from bipolar cells. If presynaptic inhibition from non-SAC amacrine cells onto bipolar cells is sufficient for direction selectivity, we would expect normal direction selectivity under this condition. In comparison, if non-SAC-mediated GABAergic inputs directly onto SACs are important, we would expect impaired direction selectivity.

We eliminated all direct GABAergic inhibition onto SACs by generating a conditional KO mouse line in which the α2 subunit of GABA$_A$ receptors, *Gabra2*, is selectively knocked out from SACs (*Gabra2* KO mice, *Figure 3A*). *Gabra2* expression colocalizes with On and Off SAC dendrites in the IPL (*Brandstätter et al., 1995*) and has been shown to be important for direction selectivity

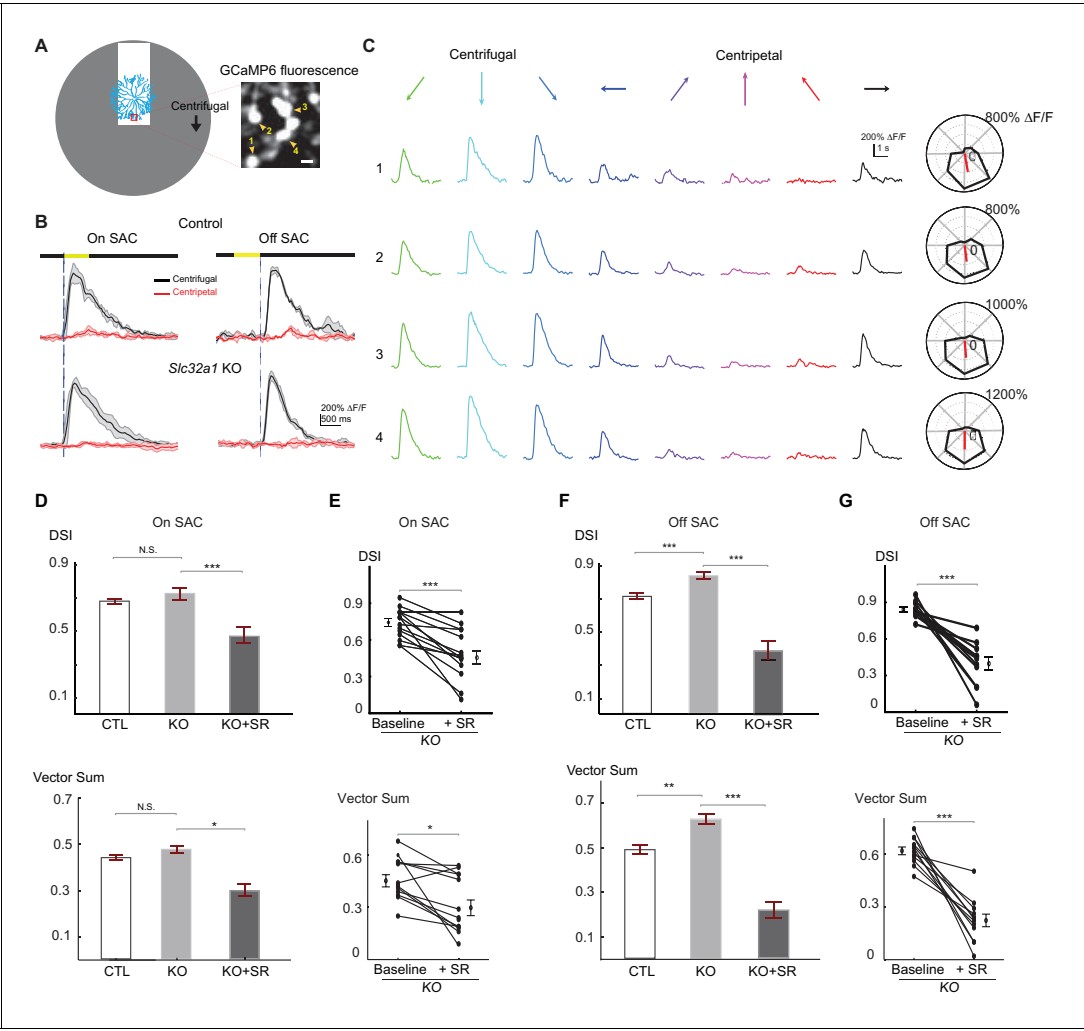

**Figure 2.** Centrifugal preference of SACs is not reduced in *Slc32a1* KO mice. (A) Left: Schematic of simple moving bar stimulus used during calcium imaging of SAC dendrites. Red square indicates the location of imaging window. Gray area indicates an area within which the moving bar (white) was presented. For the varicosities in the imaging window, downward motion of the bar corresponds to centrifugal (CF) motion, and upward motion corresponds to centripetal (CP) motion. Right: maximal intensity projection of GCaMP6m fluorescence in distal varicosities of an On-SAC from a control (C) mouse infected with AAV-floxed GCaMP6m. Four example varicosities are labeled 1–4. Scale bar: 1 μm. (B) Example GCaMP6m fluorescence traces from varicosities of the On and Off SACs in control (C) and *Slc32a1* KO (*Slc32a1* flox/flox C) mice during CF and CP motion. Dark lines represent mean values and shaded areas represent standard deviations. Yellow bar at the top indicates the time window when the bar moved across the imaging area. Blue vertical lines indicate when a leading edge reaches varicosities of On SAC and trailing edge reaches varicosities of Off SAC. (C) Example single sweeps of moving bar-evoked GCaMP6 fluorescence from varicosities 1–4 shown in (A). Polar plots of mean peak amplitude on right show centrifugal tuning of all four varicosities. (D) Summary bar graphs of DSI and vector sum values for On SACs in control (CTL) and *Slc32a1* KO mice before and after adding SR95531. CTL: DSI 0.69 ± 0.01, vector sum 0.44 ± 0.01, n = 86 cells (1221 varicosities); *Slc32a1* KO: DSI 0.73 ± 0.03, vector sum 0.48 ± 0.02, n = 20 cells (123 varicosities); *Slc32a1* KO+SR: DSI 0.43 ± 0.05, vector sum 0.3 ± 0.05, n = 14 cells (86 varicosities). DSI CTL - *Slc32a1* KO p=0.14; vector sum CTL- *Slc32a1* KO p=0.66; ***p<0.0005; *p=0.008 (E) Pairwise comparison of DSI and vector sum values of individual On-SACs before and after adding SR95531 in *Slc32a1* KO mice. Before adding SR95531: DSI 0.71 ± 0.04, vector sum 0.44 ± 0.04; After SR addition: DSI 0.43 ± 0.05, vector sum 0.3 ± 0.05. n = 14 cells (86 varicosities). ***p<0.0005; *p=0.015 (F) As in (D), summary bar graphs of DSI and vector sum values for Off-SACs. Control: DSI 0.72 ± 0.01, vector sum 0.48 ± 0.01, n = 35 cells (257 varicosities); *Slc32a1* KO: DSI 0.83 ± 0.01, vector sum 0.59 ± 0.01; n = 33 cells (353 varicosities). *Slc32a1* KO+SR: DSI 0.40 ± 0.05, vector sum 0.22 ± 0.04, n = 11 cells (178 varicosities). **p<0.005; ***p<0.0005. (G) As in (E), pairwise comparison of DSI and VS values of individual Off-SACs. Before adding SR95531: DSI 0.84 ± 0.02, vector sum 0.61 ± 0.02; After SR addition: DSI 0.4 ± 0.05, vector sum 0.22 ± 0.04. n = 11 cells (178 varicosities). ***p<0.0005. See also *Figure 2—figure supplement 1* for cumulative probability distributions of peak fluorescence in On and Off SACs of control and *Slc32a1* KO mice during CP and CF motion.

The following figure supplement is available for figure 2:

**Figure supplement 1.** Centripetal response of Off SAC dendrites is selectively reduced in Slc32a1 KO mice.

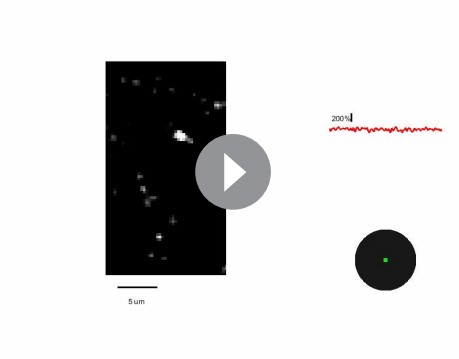

**Video 1.** Calcium imaging of On SAC varicosities expressing GCaMP6m in a control mouse in response to moving bar stimulation. This video shows continuous real-time calcium imaging of an On SAC in the control group (C) in response to a bright bar moving in eight directions. Left: raw fluorescence video, scale bar: 5 μm. Upper right: mean fluorescence change of all varicosities in the video to the left. Lower right: schematic moving bar stimulus synchronized with the imaging video and the trace. Black filled circle indicates the area of the retina where the bar was presented. Green square in the middle indicates the location of the imaging window. Calcium responses are triggered by the leading edge of the moving bar.

(*Auferkorte et al., 2012*). For targeted recording from these KOs, we also labeled SACs and a sub-type of On-Off DSGCs that prefer motion in the posterior direction (pDSGCs) (*Huberman et al., 2009*) with tdTomato and GFP, respectively (*Figure 3A*). To confirm loss of functional GABA$_A$ receptors in SACs of *Gabra2* KO mice, we recorded their spontaneous IPSCs and found that they are eliminated (*Figure 3B and C*). We also performed paired recordings from neighboring SACs, and found that none of the pairs showed evoked IPSCs (*Figure 3D*, n = 10 pairs, three mice). Therefore, GABAergic inputs onto SACs are abolished in *Gabra2* KO mice.

Although multiple studies have demonstrated that the emergence of retinal direction selectivity occurs independent of neural activity (*Elstrott et al., 2008*; *Hamby et al., 2015*; *Sun et al., 2011*; *Wei et al., 2011*), we needed to rule out potential developmental changes in this circuit in *Gabra2* KO mice. Therefore, we examined other critical synapses involved in direction selectivity: the GABAergic and cholinergic synapses between SACs and DSGCs. Using paired voltage-clamp recording, we depolarized SACs and measured evoked GABAergic IPSCs and cholinergic excitatory postsynaptic currents (EPSCs) in pDSGCs. We targeted SAC-pDSGC pairs with overlapping dendritic arbors and intersoma distances of 60–80 μm. In control mice, SACs from the null side provide strong GABAergic inputs onto pDSGCs while SACs from the preferred side provide weak GABAergic inputs. In *Gabra2* KOs, this asymmetric wiring pattern remains unchanged (*Figure 3E and F*). Similarly, cholinergic transmission in the KO is unaffected (*Figure 3E and F*). Thus our results show that GABAergic and cholinergic synapses from SACs to DSGCs are not altered in the *Gabra2* KO mice. In addition, the amplitude of EPSCs in pDSGCs evoked by a bright flashing spot is similar between control and *Gabra2* KO groups (*Figure 3G*), indicating that both cholinergic and glutamatergic synapses onto pDSGCs are unaltered.

## Centrifugal direction selectivity of off SACs is impaired in *Gabra2* KO mice during the simple moving bar stimulus

How do SACs respond to visual motion when their direct ionotropic GABAergic inhibitory inputs are lost? To address this question, we measured centrifugal direction selectivity by monitoring GCaMP6 fluorescence in On and Off SACs of *Gabra2* KO mice during the simple moving bar stimulus described above. We found that Off SACs in these KOs display a significant increase in centripetal-direction response (*Figure 4A and B*, *Video 3*), resulting in reduced

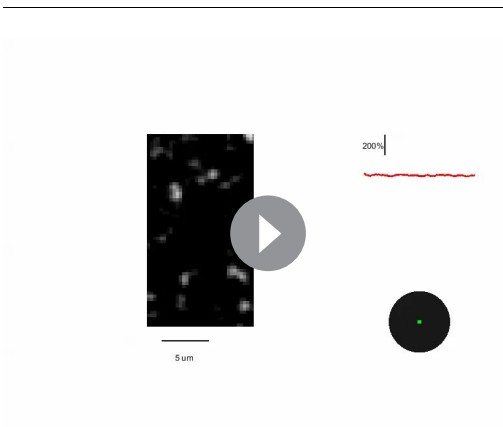

**Video 2.** Calcium imaging of Off SAC varicosities expressing GCaMP6m in a control mouse in response to moving bar stimulation. As in *Video 1*, this video shows continuous real-time calcium imaging of an Off SAC in the control group in response to the moving bar stimulus. Note that calcium responses are triggered by the trailing edge of the moving bar.

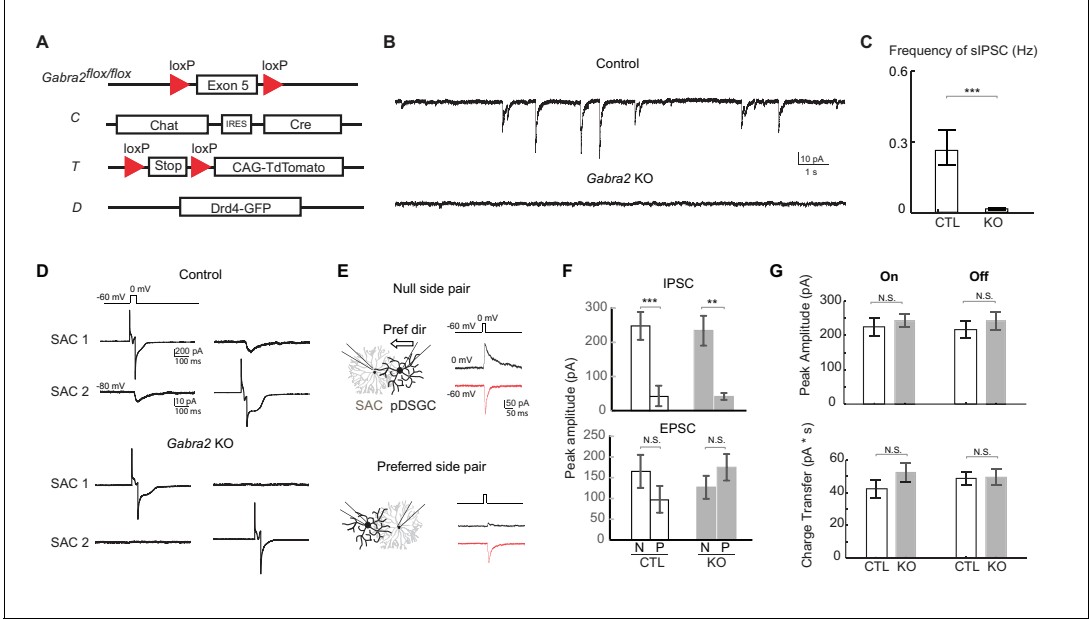

**Figure 3.** Genetic deletion of *Gabra2* from SACs eliminates GABAergic inputs onto SACs without affecting the synapses between SACs and pDSGCs. (**A**) Schematic diagram shows transgenes involved in *Gabra2* conditional KO mice. The KO mice carry homozygous floxed *Gabra2* allele to replace the endogenous *Gabra2* gene for Cre-dependent excision (acronym: *Gabra2*$^{flox/flox}$), and *Chat-IRES-Cre* (**C**). The floxed *tdTomato* (**T**) and a *Drd4-GFP* transgene to label pDSGCs (**D**) were also included when SACs or pDSGCs were targeted for electrophysiological recordings. (**B**) Example traces of spontaneous IPSCs (sIPSCs) in On SACs from control (*CT*) and *Gabra2* KO (*Gabra2*$^{flox/flox}$ *CT*) mice. (**C**) Summary bar graph of sIPSC frequency in control and KO groups. Control: 0.28 ± 0.07 Hz, n = 18 cells, five mice; KO: 0.02 ± 0.04 Hz, n = 16 cells, four mice. ***p<0.0005. (**D**) Example evoked IPSC traces from reciprocal On SAC-SAC pairs in Control and *Gabra2* KO mice. (**E**) Voltage clamp traces from null-side and preferred-side SAC-DSGC pairs in control (*CTD*) and KO (*Gabra2*$^{flox/flox}$*CTD*) mice showing cholinergic EPSCs (red inward) and GABAergic IPSCs (black outward) evoked in pDSGCs by depolarizing SACs. Schematic on left shows soma locations of the null and preferred side pairs. Black arrows indicate the pDSGC's preferred direction. (**F**) Summary bar graphs of IPSC and EPSC peak amplitudes in pDSGCs evoked by null (N) and preferred (P) side SACs in control and KO groups. IPSC: Control – N: 247.9 ± 40.8 pA, n = 14 cells, seven mice; *Control* – P: 43.0 ± 30.1 pA, n = 14 cells, seven mice; KO – N: 233.8 ± 43.7 pA, n = 7 cells, four mice; KO – P: 41.3 ± 10.0 pA, n = 8 cells, five mice, **p<0.005, ***p<0.0005; EPSC: *Control* – N: 165.2 ± 40.4 pA, n = 14 cells, seven mice; *Control* – P: 96.6 ± 32.2 pA, n = 14 cells, seven mice; KO – N: 127.4 ± 20.4 pA, n = 7 cells, four mice; KO – P: 175.0 ± 31.9 pA, n = 8 cells, five mice, one way ANOVA p=0.07. (**G**) Summary bar graphs of EPSC amplitudes and total charger transfer in pDSGCs evoked by onset and offset of a bright spot in control and KO mice. Peak amplitude: Control On, 224.6 ± 26.7 pA; KO On, 242.7 ± 19.5 pA, p=0.58; Control Off, 216.5 ± 23.8 pA; KO Off, 242.4 ± 25.9 pA, p=0.46. Charge Transfer: Control On, 42.3 ± 5.6 pA•s; KO On, 52.2 ± 5.9 pA•s, p=0.22; Control Off, 48.7 ± 4.2 pA•s; KO Off, 49.4 ± 4.9 pA•s, p=0.92. Control, n = 21 cells, nine mice; KO, n = 30 cells, 11 mice.

direction selectivity (*Figure 4C–4F*). Therefore, direct GABAergic inputs onto Off SACs from non-SAC amacrine cells are important for centrifugal direction selectivity. In contrast, we found that On SACs in the KOs exhibit centrifugal preference similar to those of control mice (*Figure 4*, *Video 4*), suggesting that direct GABAergic inputs are not required for direction selectivity in On SACs during the simple moving bar stimulus. Bath application of SR95531 reduced direction selectivity of On SACs in KOs (*Figure 4—figure supplement 1*), suggesting a role for presynaptic inhibition onto bipolar cells in generating direction selectivity.

## Off inhibitory inputs onto On-Off DSGCs display impaired direction selectivity in *Gabra2* KO mice

Since centrifugal preference of Off SAC dendrites is selectively impaired in *Gabra2* KO mice, we would expect that the Off component of inhibitory inputs onto DSGCs would also display reduced direction selectivity. We tested this hypothesis by performing whole-cell voltage clamp recordings in pDSGCs to measure light-evoked IPSCs during the simple moving bar stimulus. To quantify direction selectivity of the inhibitory inputs onto DSGCs, we calculated DSI and vector sum of the peak amplitude and total charge transfer of IPSCs in response to the simple moving bar stimulus. In control mice, both On and Off components of IPSCs are direction selective (*Figure 5A–5D*). In *Gabra2* KO

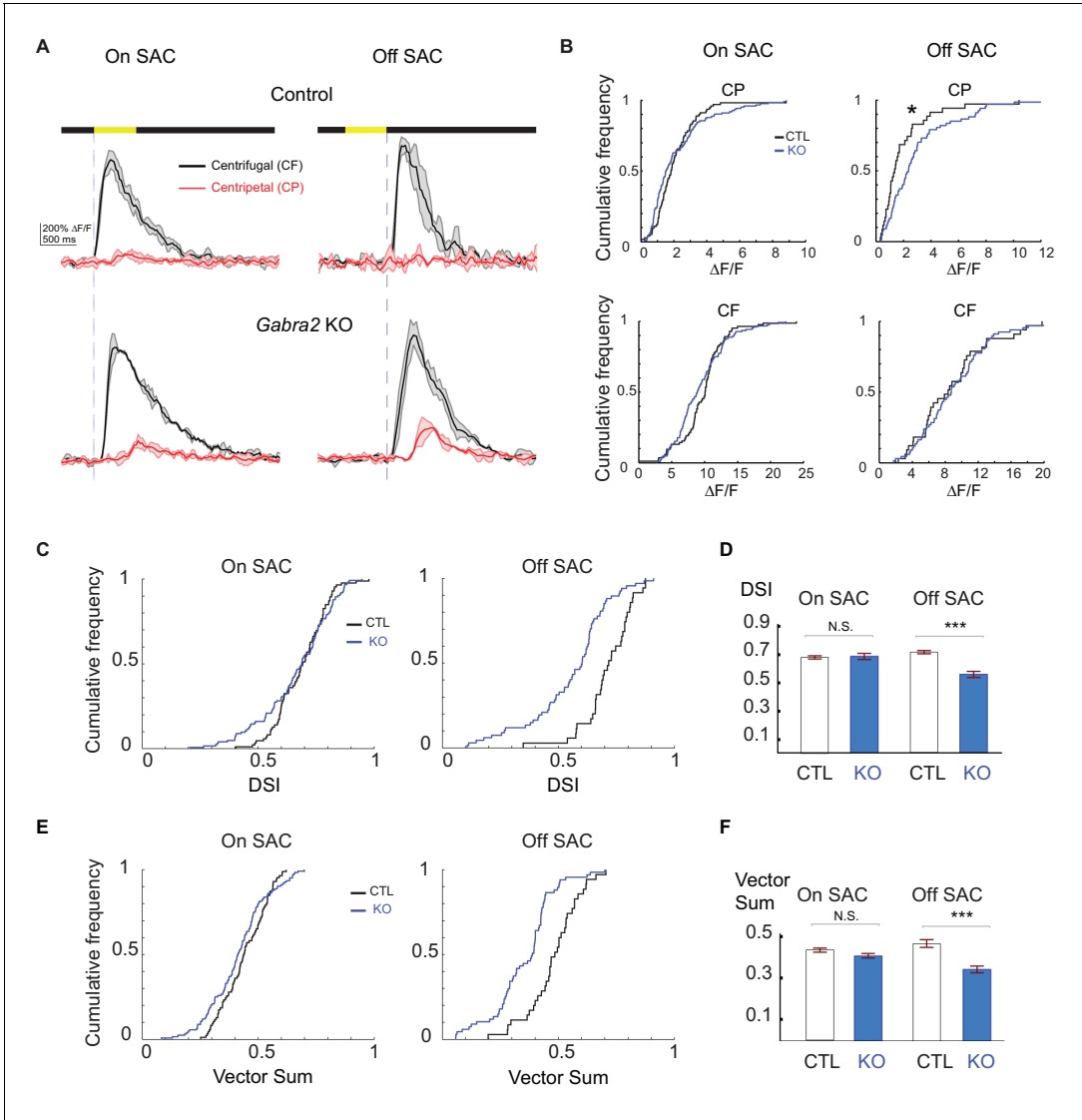

**Figure 4.** Centrifugal preference of Off, but not On, SACs is impaired in *Gabra2* KO mice. (A) Example GCaMP6m fluorescence traces from varicosities of On (left) and Off (right) SACs in control (C) and KO (*Gabra2^{flox/flox} C*) mice during CF and CP motion. Dark lines represent mean values and shaded areas represent standard deviations. Yellow bar at the top indicates the time window when the bar moved across the imaging area. Blue vertical lines indicate when a leading edge reaches varicosities of On SAC and trailing edge reaches varicosities of Off SACs. (B) Cumulative probability distributions of peak fluorescence from dendritic varicosities in On (left) and Off (right) SACs of control and *Gabra2* KO mice during CP and CF motion. (C) Cumulative probability distributions of DSI values from dendritic varicosities in On and Off SACs of control and KO mice. (D) Summary bar graph of DSI values from dendritic varicosities in On and Off SACs of control and KO mice: On SAC: Control 0.69 ± 0.01, KO 0.71 ± 0.02, p=0.22; Off SAC: Control 0.72 ± 0.01, KO 0.55 ± 0.02, ***p<0.0005. (E) Cumulative probability distributions of vector sum values from dendritic varicosities in On and Off SACs of control and KO mice. (F) Summary bar graph of vector sum values from dendritic varicosities in On and Off SACs of control and KO mice. On SAC: control 0.44 ± 0.01, KO 0.41 ± 0.01, p=0.26; Off SAC: control 0.48 ± 0.02, KO 0.35 ± 0.02, ***p<0.0005. Control On n = 86 cells (1221 varicosities); KO On n = 124 cells (1467 varicosities); Control Off n = 35 cells (257 varicosities); KO Off n = 67 cells (897 varicosities). See also *Figure 4—figure supplement 1* for the effect of SR95531 on direction selectivity of On SACs in *Gabra2* KO mice.

The following figure supplement is available for figure 4:

**Figure supplement 1.** Centrifugal preference of On SACs is impaired in Gabra2 KO mice in the presence of SR95531.

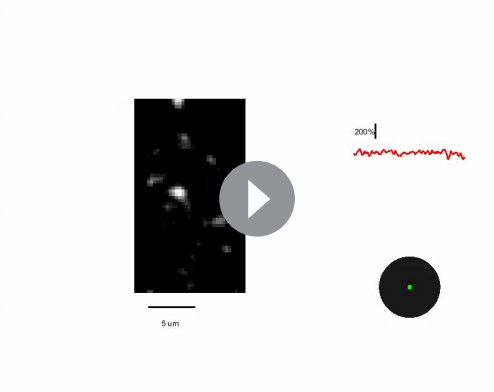

**Video 3.** Calcium imaging of Off SAC varicosities expressing GCaMP6m in a *Gabra2* KO mouse in response to moving bar stimulation. As in *Video 1*, this video shows continuous real-time calcium imaging of an Off SAC in the *Gabra2* KO group (*Gabra2*^*flox/flox* *C*) in response to the moving bar stimulus.

mice, the Off component of IPSCs shows reduced selectivity (*Figure 5C and D*) but the On component does not (*Figure 5A and B*). Therefore, in the absence of GABAergic inputs onto SACs, the Off component of inhibitory inputs onto DSGCs displays impaired directional tuning.

## Off component of On-Off DSGC spiking activity shows impaired direction selectivity in *Gabra2* KO mice

To test if loss of lateral inhibition onto Off SACs impacts the ultimate output of the retinal direction selective circuit, we determined its effect on

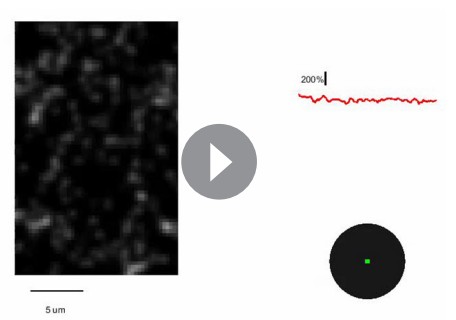

**Video 4.** Calcium imaging of On SAC varicosities expressing GCaMP6m in a *Gabra2* KO mouse in response to moving bar stimulation. As in *Video 1*, this video shows continuous real-time calcium imaging of an On SAC in the *Gabra2* KO group in response to the moving bar stimulus.

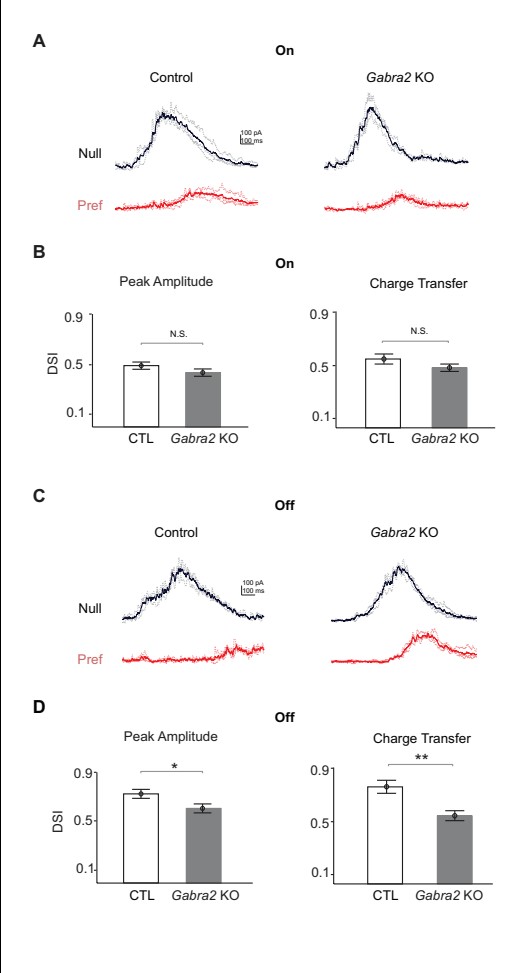

**Figure 5.** Off inhibitory inputs onto pDSGCs display impaired directional selectivity in *Gabra2* KO mice. (**A**) Example IPSC traces of pDSGCs evoked by leading edge (On) of a bright bar moving in the preferred and null directions in control (*CTD*) and *Gabra2* KO (*Gabra2*^*flox/flox* *CTD*) mice. (**B**) Summary bar graphs of DSI values for peak amplitude and charge transfer of the On component of IPSCs in control and KO groups. Peak Amplitude: Control 0.49 ± 0.03, KO 0.44 ± 0.03, p=0.19; Charge Transfer: Control 0.53 ± 0.04, KO 0.49 ± 0.03, p=0.16. Control n = 32 cells, 13 mice; KO n = 33 cells, 18 mice. (**C**) As in (**A**), example IPSC traces of pDSGCs evoked by trailing edge (Off) of a bright moving bar. (**D**) As in (**B**), summary bar graphs for the Off component of IPSCs in pDSGCs. Control 0.72 ± 0.03, KO 0.58 ± 0.03, *p=0.01; Charge Transfer: Control 0.73 ± 0.03, KO 0.51 ± 0.03, **p<0.005.

the spiking activity of DSGCs. We recorded the spiking activity of pDSGCs in loose-patch configuration, and compared the direction selectivity of the leading edge-evoked On response and the trailing edge-evoked Off response in control and *Gabra2* KO mice during the simple moving

bar stimulus. Consistent with the reduced directional tuning of IPSCs observed in the Off component, we found that the Off response of pDSGC spiking activity in *Gabra2* KO mice shows less direction selectivity than that in control mice (*Figure 6A and C*). In contrast, the On responses in KOs display direction selectivity similar to that of controls (*Figure 6A and B*). To rule out the possibility that the Off response is influenced by prior activation of the On pathway, we reversed the contrast of the moving bar stimulus by presenting a dark bar against a bright background. We found that direction selectivity of the leading edge-evoked Off component in *Gabra2* KOs was again less than that in controls (*Figure 6D and F*), while the trailing edge-evoked On response of KOs shows direction selectivity similar to controls (*Figure 6D and E*). Therefore, the Off component of pDSGC spiking activity becomes less directionally tuned in *Gabra2* KO mice.

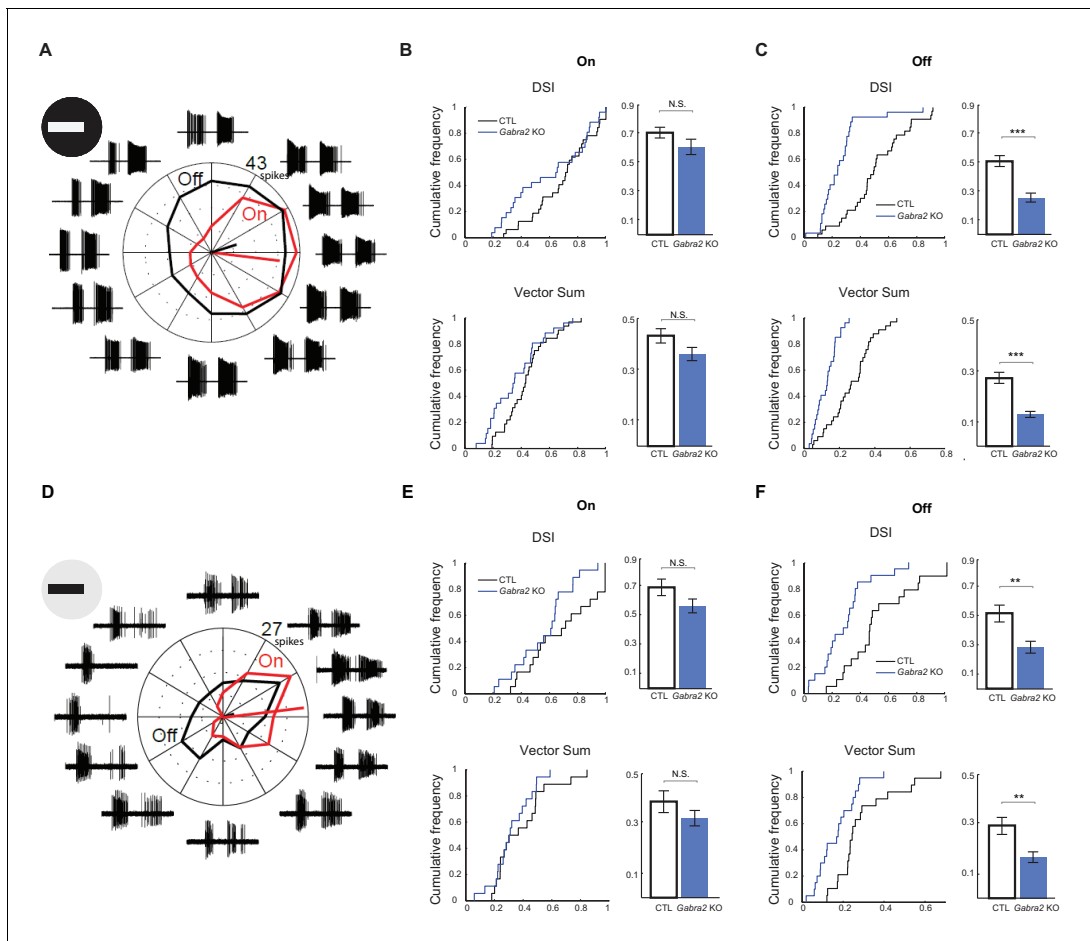

**Figure 6.** Off component of pDSGC spiking activity displays impaired directional tuning in *Gabra2* KO mice. (**A – C**) Spiking response of pDSGCs to a bright moving bar. Control n = 33 cells, 20 mice; *Gabra2* KO n = 37 cells, 14 mice. (**A**) Example loose-patch recordings and polar plot for a pDSGC in a *Gabra2* KO (*Gabra2^flox/flox CTD*) mouse in response to a bright bar moving in 12 directions. Trace for each direction is the overlay of three trials. The leading edge-evoked On and trailing edge-evoked Off components of the spiking activity can be clearly separated. Mean spike counts for On (red) and Off (black) responses are shown on the polar plot. (**B**) Cumulative distributions (left) and summary bar graphs (right) of DSI and vector sum values for On responses of pDSGCs during the bright moving bar stimulus. DSI: Control 0.70 ± 0.03, KO 0.60 ± 0.04, p=0.14; vector sum: Control 0.43 ± 0.03; KO 0.36 ± 0.03, p=0.08. Control n = 59 cells, 20 mice; KO n = 46 cells, 14 mice. (**C**) As in (**B**), DSI and vector sum for Off responses of pDSGCs during the bright moving bar stimulus. DSI: Control 0.51 ± 0.04, KO 0.26 ± 0.03, ***p<0.0005; vector sum: Control 0.27 ± 0.02, KO 0.13 ± 0.01, ***p<0.0005. (**D – F**) Spiking response of pDSGCs to a dark moving bar. Control n = 19 cells, nine mice; KO n = 20 cells, nine mice. (**D**) As in (**A**), example loose-patch recordings and polar plot for a pDSGC in a *Gabra2* KO (*Gabra2^flox/flox CTD*) mouse in response to a dark bar moving in 12 directions. (**E**) As in (**B**), DSI and vector sum for On responses of pDSGCs during the dark moving bar stimulus. DSI: Control 0.69 ± 0.05, KO 0.57 ± 0.04, p=0.09; vector sum: Control 0.39 ± 0.04, KO 0.32 ± 0.03, p=0.23. Control n = 19 cells, nine mice; KO n = 20 cells, nine mice. (**F**) As in (**C**), DSI and vector sum for Off responses of pDSGCs during the dark moving bar stimulus. DSI: Control 0.52 ± 0.05, KO 0.28 ± 0.04, **p<0.005; VS: Control 0.29 ± 0.03, KO 0.16 ± 0.02, **p<0.005.

## Lateral inhibition in the On pathway is required for direction selectivity on a noisy background

In the experiments above, we used a simple moving bar stimulus and found that direction selectivity of the Off pathway requires lateral inhibition onto Off SACs, but direction selectivity of the On pathway does not require direct GABAergic input onto On SACs. What, then, is the role of lateral inhibition in the On pathway? We hypothesized that our stimulus conditions did not optimally activate this inhibitory pathway. In particular, a noisy background may more effectively engage this inhibition and reveal its functional importance in direction selectivity. To test this idea, we measured the direction selectivity of pDSGCs in *Gabra2* KO mice using a bright bar moving against a flickering checkerboard background. This flickering checkerboard provides "white noise" that is commonly used for mapping receptive fields in the spike-triggered average method (*Chichilnisky, 2001*). We set the brightness of this moving bar equal to that of our simple moving bar stimulus, and varied the brightness of the checker pattern from 0% to 100% of the moving bar intensity. In control mice, the pDSGC spiking response maintains direction selectivity over a wide range of checker intensities (*Figure 7A–7C*). Consistent with the earlier finding that lateral inhibition is crucial for direction selectivity of the Off component (*Figure 6C and E*), the direction selectivity of the Off response in *Gabra2* KO mice is reduced regardless of checker intensity (*Figure 7A and C*). Interestingly, however, replacing the gray background with a flickering checkerboard also significantly reduces the On direction selectivity in KO mice (*Figure 7A and B*), indicating a functional contribution of lateral inhibition during more complex stimuli.

The flickering checkerboard not only adds noise to the visual stimuli, but also changes the average background illuminance and therefore the effective contrast of the moving bar. To test if changing either ambient illuminance or contrast alone can mimic the effect of introducing the flickering checkerboard, we measured the direction selectivity of pDSGCs in *Gabra2* KO mice using the simple moving bar stimulus with different contrast and background illuminance levels that lie in the upper and lower range of the flickering checkerboard stimuli. We found that, similar to the results with our original simple bar stimulus (*Figure 6*), the On component of the pDSGC's spiking response displays normal direction selectivity, while the Off component shows reduced direction selectivity at all illuminance and contrast levels tested (*Figure 7D*). Therefore, the impaired direction selectivity seen in the On pathway in *Gabra2* KO mice with the flickering checkboard background is not due to changes in stimulus contrast or ambient illuminance. Instead, the noise introduced by the flickering checkerboard is important for the functional recruitment of lateral inhibition in the On pathway.

## Discussion

Here we used synapse-specific genetic manipulations to distinguish the effects on the retinal direction selectivity of GABAergic inhibition originating from different sources. We found that distinct sets of inhibitory mechanisms are recruited, depending on the properties of the background and the positive versus negative contrast of the moving object. The optimal direction selectivity of a dark object, encoded by the Off pathway, requires lateral inhibition from non-SAC amacrine cells onto Off SACs. In contrast, direction selectivity of a bright object, encoded by the On pathway, does not require lateral inhibition onto On SACs when the moving bar stimulus lies on a homogeneous background, but does require it when the bar moves on a noisy background. Together, our results highlight the multiple levels of synaptic mechanisms that underlie direction selectivity under complex visual conditions.

### *Slc32a1* KO and *Gabra2* KO mice display no detectable developmental compensations

Although using conditional KO mice offers the unique advantage of achieving the cell type and synapse specificity, a major concern of this genetic approach is that neural circuits may potentially be altered due to developmental compensations. In our study, we selectively remove from SACs the two genes involved in GABAergic transmission, *Slc32a1* and *Gabra2*, during the early postnatal period (*Xu et al., 2016*). Therefore, we must verify that aside from loss of SAC-mediated GABA release in *Slc32a1* KO mice and loss of GABA$_A$ receptors in SACs in *Gabra2* KO mice, the rest of the direction selectivity circuit is not altered. To address this issue, we examined the other known

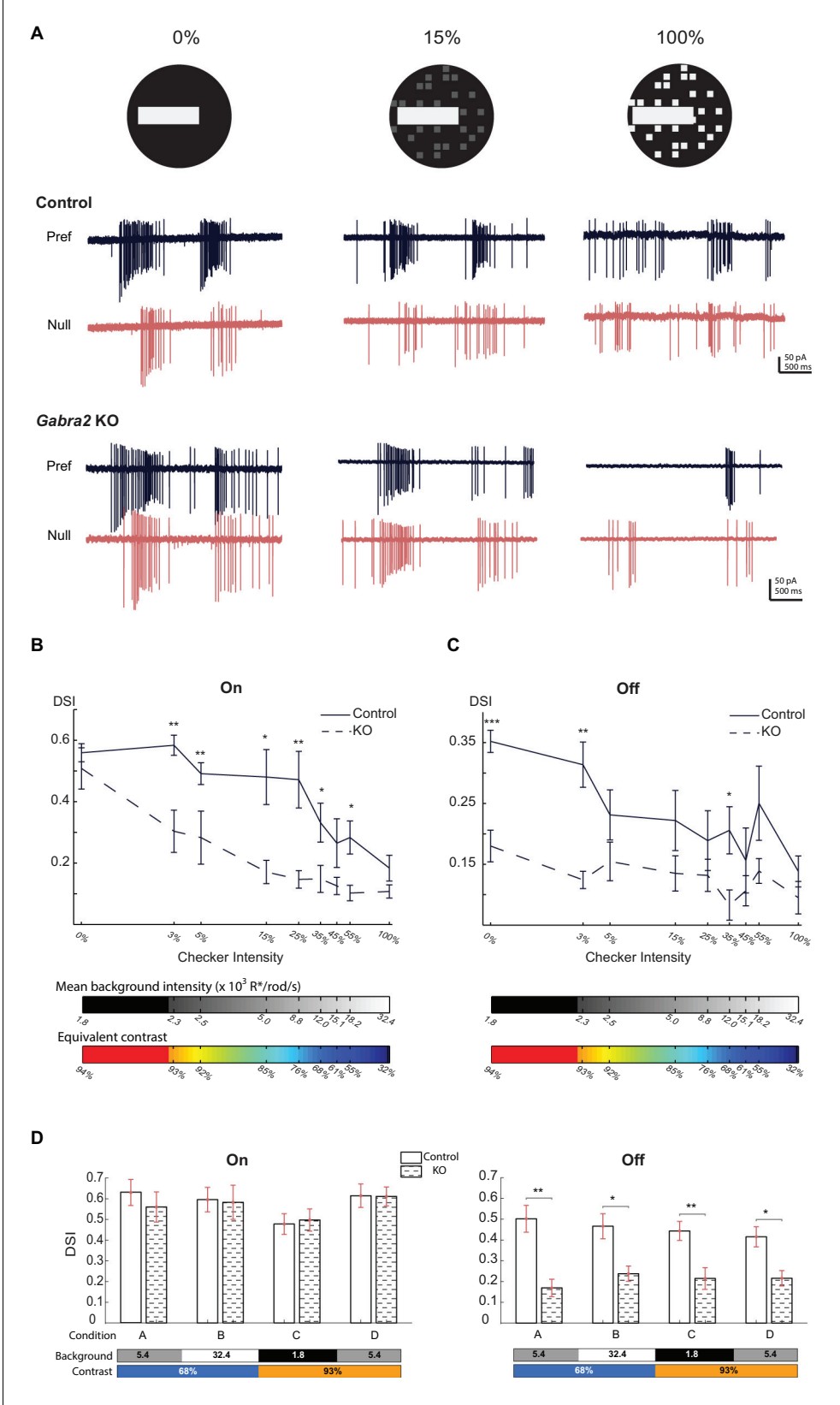

**Figure 7.** Direct inhibitory inputs onto On SACs are required for robust direction selectivity against noisy background. (**A**) Visual stimulus (top) is a bright bar moving on a checkered background with different checker intensities, and loose-patch recordings (bottom) show pDSGC responses in control (*CTD*) and *Gabra2* KO

*Figure 7 continued on next page*

*Figure 7 continued*

(*Gabra2^flox/flox CTD*) mice in preferred and null directions. (**B**) Summary plot of DSI values of the On component of pDSGC spiking activity evoked by a bright bar moving against a flickering checkerboard background. Checker intensity is expressed as the percentage of the brightness of the moving bar. Bars below plot indicate the mean background intensity of the flickering checkerboard (top) and the equivalent contrast of the moving bar stimulus (bottom). Control, n = 12 cells, four mice; *Gabra2* KO, n = 18 cells, four mice. (**C**) As in (**B**), summary plot of DSI of Off component of pDSGC spiking activity. (**D**) Summary bar graphs of DSI values from On and Off components of pDSGC spiking at different background illuminance and contrast levels during the simple moving bar stimulus against a homogeneous background. Bars below plots indicate the background intensity (x10$^3$ R*/rod/s) and contrast of the bar. On: Condition A: Control 0.63 ± 0.06, n = 17 cells from seven mice, KO 0.56 ± 0.07, n = 14 cells from six mice, p=0.31; Condition B: Control 0.60 ± 0.06, n = 8 cells from five mice, KO 0.58 ± 0.08, n = 10 cells from five mice, p=0.83; Condition C: Control 0.48 ± 0.05, n = 12 cells from five mice, KO 0.49 ± 0.05, n = 10 cells from five mice, p=0.95; Condition D: Control 0.61 ± 0.06, n = 14 cells from seven mice, KO 0.61 ± 0.04, n = 12 cells from five mice, p=0.73; Off: Condition A: Control 0.50 ± 0.06, KO 0.17 ± 0.04, p=0.004; Condition B: Control 0.47 ± 0.06, KO 0.24 ± 0.04, p=0.01; Condition C: Control 0.44 ± 0.05, KO 0.21 ± 0.05, p=0.004; Condition D: Control 0.42 ± 0.05, KO 0.22 ± 0.04, p=0.006. For (**B-D**), *p<0.05; **p<0.005; ***p<0.0005.

synapse types that are involved in retinal direction selectivity, including the cholinergic and GABAergic synapses between SACs and DSGCs, as well as the glutamatergic inputs onto DSGCs from bipolar cells. We found that these synapses showed no detectable changes (*Pei et al., 2015*) and *Figure 3*), indicating that removing GABA release or GABA receptors from SACs does not lead to compensatory developmental changes at other synapses in the direction selective circuit. We cannot discount the possibility that developmental compensation has occurred at sites not examined in this study. However, our findings are consistent with multiple previous studies that demonstrate that the early development of retinal direction selectivity is highly resistant to changes in neural activity (*Chan and Chiao, 2008*; *Elstrott et al., 2008*; *Hamby et al., 2015*; *Sun et al., 2011*; *Wei et al., 2011*).

After direction selectivity is established, visual experience has been shown to play a role in refining the clustered distribution of preferred directions of On-Off DSGCs (*Bos et al., 2016*). In both *Slc32a1* KO and *Gabra2* KO mice, visually evoked responses are present in all retinal neurons including SACs and pDSGCs. Furthermore, pDSGCs in both KO lines remain tuned to the posterior direction (*Pei et al., 2015*); data not shown). Given that all known synaptic connections in the direction selective circuit and their strengths are preserved in our conditional KO mice, we consider these mouse lines valuable tools for circuit analysis and for testing existing models.

## The role of lateral inhibition with a simple moving bar stimulus

### The on pathway

During object motion in the centripetal direction, lateral inhibition is thought to selectively suppress SAC dendritic activation via inhibitory inputs from the surround (*Lee and Zhou, 2006*). A major candidate source for this lateral inhibition is GABAergic input from neighboring SACs (*Lee and Zhou, 2006*). However, when a bar moves against a homogeneous dark background, we found that blocking GABA release or GABA receptors of On SACs does not affect their centrifugal preference. Therefore, under this stimulus condition, lateral inhibition is not necessary for centrifugal direction selectivity of On SACs, and other mechanisms (*Hausselt et al., 2007*; *Kim et al., 2014*; *Oesch and Taylor, 2010*; *Vlasits et al., 2016*) must act to maintain it.

### The off pathway

Unlike On SACs, Off SACs are sensitive to lateral inhibition during the simple bar stimulus. Removing all GABAergic inputs onto Off SACs in *Gabra2* KO mice reduces centrifugal preference. As a result, the Off component of pDSGC inhibitory inputs and spiking responses shows significantly impaired direction selectivity. This effect does not stem from SAC-SAC inhibition, since the loss of this inhibition in *Slc32a1 KO* mice does not reduce direction selectivity of Off SACs, but rather slightly enhances it by decreasing centripetal-direction response. This indicates that the centrifugal preference of Off SAC dendrites requires additional GABAergic inputs onto Off SACs from non-SAC amacrine

cells. These cells are presumably the wide-field amacrine cells that synapse onto the proximal dendrites of Off SACs and contribute about 8% of the total number of inhibitory synapses onto Off SACs identified by connectomic reconstruction (*Ding et al., 2016*). Though these synapses represent a small fraction of the synaptic inputs onto Off SACs, they may be sufficiently strong and efficient for shunting glutamatergic excitatory inputs that are also clustered in the proximal SAC dendrites (*Vlasits et al., 2016*).

We still do not know what mechanism underlies the improved selectivity of Off SACs in *Slc32a1* KO mice, and if this improvement detected by calcium imaging impacts the spiking output of On-Off DSGCs. One possibility is that Off SACs also provide GABAergic inhibition to the above-mentioned non-SAC amacrine cells. In the absence of GABA release from SACs in *Slc32a1* KO mice, these non-SAC amacrine cells might be disinhibited, thus providing stronger lateral inhibition to Off SACs to decrease their centripetal-direction response. Directly testing this hypothesis requires targeted recordings from pairs of Off SACs and non-SAC amacrine cells, an experiment that is currently hindered by a lack of cell type-specific markers.

We have shown here that the direction selectivity of On and Off SACs is differentially affected in both *Slc32a1* KO and *Gabra2* KO mice. Thus, the direction selective circuitry is not merely mirrored in the On and Off layers of the IPL, but consists of distinct sets of mechanisms in each sublamina. Our study therefore adds to others that have demonstrated differences and interactions between the On and Off pathways of the direction selective circuit (*Ackert et al., 2009*; *Taylor and Vaney, 2002*; *Vlasits et al., 2014*) and of other retinal circuits (*Geffen et al., 2007*; *Tikidji-Hamburyan et al., 2015*). Moreover, our study shows that GABAergic inhibition onto SACs consists of multiple facets that are engaged differently during retinal processing of positive- and negative-contrast motion.

## The role of lateral inhibition during a moving bar stimulus against a noisy background

Lateral inhibition plays an important role in feature selection in multiple sensory systems. One elaborated form, the reciprocal inhibition between inhibitory neurons, has been described in multiple brain areas such as retina, thalamus, midbrain and cortex (*Lee and Zhou, 2006*; *Machens et al., 2005*; *Miller and Wang, 2006*; *Papadopoulou et al., 2011*). In sensory and cognitive systems, lateral inhibition motifs have been implicated in selecting the target feature in an environment when other 'competitor' features are present (*Mysore and Knudsen, 2012*; *Sharpee, 2012*). In the visual system, moving objects in natural scenes have backgrounds that are rarely homogeneous and stationary, but rather are spatially and temporally noisy. These can serve as 'competitor' stimuli that pose challenges to the direction selective circuit for reliably detecting motion direction. We postulate that lateral inhibition onto On SACs functions to safeguard direction selectivity against a noisy environment. A noisy background increases activation of bipolar cells, which in turn provides stronger glutamatergic drive to the downstream amacrine cells including SACs. Under this condition, the lateral inhibitory network may be in a more 'primed' state to be efficiently recruited by the motion stimuli and to exert its function. In contrast, amacrine cells involved in lateral inhibition are less activated when bipolar cells adapt to the gray background, and thus the moving bar alone may not be sufficient to significantly recruit lateral inhibition in the On pathway. Consistent with this hypothesis, direction selectivity of the On response of DSGCs in *Gabra2* KO mice is normal when the background is homogeneous but deteriorates as soon as white noise is introduced into the background. These findings represent an intriguing example in which additional neural mechanisms are recruited when the visual stimulus more closely resembles natural viewing conditions, but simpler visual stimuli involving only the feature of interest may not reveal the functional significance of these mechanisms (*David et al., 2004*; *Felsen and Dan, 2005*; *Felsen et al., 2005*; *Turner and Rieke, 2016*). A notable analogy has been reported in the visual cortex: a prominent inhibitory component in the receptive field of visual cortical neurons is uniquely revealed by more complex natural stimuli but is not observed with synthetic sinusoidal gratings (*David et al., 2004*).

## Comparison to previous pharmacological results using GABA receptor antagonist

The role of lateral inhibition onto SACs in direction selectivity has been exclusively studied in On SACs using bath application of the $GABA_A$ receptor antagonist SR95531 with variable results (*Ding et al., 2016*; *Euler et al., 2002*; *Gavrikov et al., 2006*; *Hausselt et al., 2007*; *Lee and Zhou, 2006*; *Oesch and Taylor, 2010*; *Vlasits et al., 2016*). This variability may be due to a number of factors, such as species differences, conditions of visual stimulation, somatic recordings versus calcium imaging at the dendrites, and locations of varicosities along SAC dendrites. Furthermore, the effect of the antagonist cannot be attributed to specific type(s) of GABAergic synapses since $GABA_A$ receptors are extensively expressed in amacrine cells and bipolar cells in the IPL. Here, we combined pharmacology with the genetic dissection of the GABAergic circuitry to determine the synaptic loci that participate in direction selectivity. This approach allowed us to assign functional roles to different inhibitory synaptic components, and to identify a new component in the functional wiring diagram of the direction selective circuit: inhibitory inputs from non-SAC amacrine cells onto Off SACs.

Pharmacological experiments cannot establish the ultimate effect that lateral inhibition onto SACs has on the spiking output of DSGCs because SR95531 disrupts SAC-DSGC inhibition. However, by genetically eliminating $GABA_A$ receptors but not neurotransmitter release from SACs in *Gabra2* KO mice, we preserved the critical synaptic connections between SACs and DSGCs, and determined the role that GABAergic inputs onto SACs play in the spiking output of DSGC's. We found that the direction selectivity of On-Off DSGC spiking decreases when SAC centrifugal preference is reduced. Therefore, the inhibitory mechanisms contributing to the centrifugal preference of SACs that we have studied here are functionally relevant for determining the signals that are relayed from the retina to the brain.

## Materials and methods

### Mice

The *Gabra2*^flox/flox^ mouse line was a generous gift from Dr. Uwe Rudolph at Harvard Medical School. *Slc32a1*^flox/flox^ mice (*Slc32a1<tm1Lowl>/J*), *Chat-IRES-Cre* mice (129S6-*Chat*^tm2(cre)Lowl^/J) and floxed *tdTomato* mice (129S6-Gt(ROSA)26Sor^tm9(CAG-tdTomato)Hze^/J) were acquired from the Jackson Laboratory. *Drd4–GFP* mice were originally developed by MMRRC (http://www.mmrrc.org/strains/231/0231.html) in the Swiss Webster background, and were subsequently backcrossed to C57BL/6 background. All strains were backcrossed to the *C57BL/6* background in our laboratory, and crossed to each other to create the lines used in this study. Mice of both sexes between postnatal days 18–35 were used for paired recording experiments, and those between postnatal days 24–35 were used for light response experiments. All procedures to maintain and use mice were in accordance with the University of Chicago Institutional Animal Care and Use Committee (Protocol number ACUP 72247) and in conformance with the NIH Guide for the Care and Use of Laboratory Animals and the Public Health Service Policy.

### Preparation of isolated retina

Mice were anaesthetized with isoflurane and decapitated after dark adaptation. Under infrared illumination, retinas were isolated from the pigment epithelium at room temperature in oxygenated Ames' medium (Sigma-Aldrich, St. Louis, MO) for visual stimulation experiments or in artificial cerebrospinal fluid (ACSF) containing 119.0 mM NaCl, 26.2 mM $NaHCO_3$, 11 mM D-glucose, 2.5 mM KCl, 1.0 mM $K_2HPO_4$, 2.5 mM $CaCl_2$, and 1.3 mM $MgCl_2$ for dual patch clamp recording. Isolated retinas were then cut into dorsal or ventral halves and mounted ganglion-cell-layer-up on top of a 1 $mm^2$ hole in a small piece of filter paper (Millipore, Billerica, MA). The orientation of the preferred direction (posterior) *of Drd4-GFP* positive neurons was noted for each piece. Retinas were kept in darkness at room temperature in Ames' medium or ACSF bubbled with 95% O2/5% $CO^2$ until use (0–7 hr).

### Whole-cell voltage-clamp recording

Recording electrodes of 3–5 MΩ were filled with a cesium-based internal solution containing 110 mM $CsMeSO_4$, 2.8 mM NaCl, 4 mM EGTA, 5 mM TEA-Cl, 4 mM adenosine 5'-triphosphate

(magnesium salt), 0.3 mM guanosine 5′-triphosphate (trisodium salt), 20 mM HEPES, 10 mM phosphocreatine (disodium salt), 5 mM N-Ethyllidocaine chloride (QX314), 0.025 mM Alexa 488 (for SACs) and 0.025 mM Alexa 594 (for pDSGCs), pH 7.25. tdTomato-labeled SACs and GFP-labeled pDSGCs were identified with epifluorescence imaging (X-Cite) or two-photon microscopy (Bruker Nano Surfaces Division, Middleton, WI) under a water immersion objective (60x, Olympus LUM-PlanFl/IR). Data were acquired using PCLAMP 10 recording software and a Multiclamp 700B amplifier (Molecular Devices, Sunnyvale, CA), low-pass filtered at 4 kHz and digitized at a sampling rate of 10 kHz. Light-evoked responses were recorded at a bath temperature of 32–33°C. Spontaneous IPSCs in SACs and paired SAC-SAC and SAC-pDSGC recordings were obtained at room temperature.

During paired SAC-pDSGC recordings, the evoked IPSCs and EPSCs in pDSGCs were isolated by holding the cells at reversal potentials for these conductances (0 mV for GABAergic, −60 mV for cholinergic) in the presence of 0.05 mM D-AP5 and 0.05 mM DNQX disodium salt. The reversal potentials were calculated and then experimentally confirmed by obtaining current-voltage (I-V) relationships of pharmacologically isolated GABAergic and cholinergic currents in pDSGCs. To measure spontaneous IPSCs in SACs and evoked IPSCs in SAC-SAC paired recordings, 0.008 mM DHβE, 0.05 mM D-AP5 and 0.05 mM DNQX disodium salt were bath applied to the retinas to block NMDA-, AMPA/kainite- and nicotinic receptors. Since depolarizing SACs to 0 mV caused poorly controlled voltage-gated conductances at the beginning of the depolarizing step, we clamped the postsynaptic SACs at −80 mV to measure inward chloride currents while all excitatory channels were pharmacologically blocked. Reported potentials have been corrected for the liquid junction potential (~9 mV).

## Analysis of whole-cell voltage clamp data

Light-evoked IPSCs in pDSGCs were isolated by holding cells at 0 mV. Three repetitions of raw synaptic traces were recorded and averaged to obtain the mean response for each stimulus condition. The peak amplitude and total charge transfer of IPSCs evoked by the leading and trailing edges of the moving bar were used to calculate the direction selective index (DSI) and vector sum. DSI is defined as $\left|\frac{P-N}{P+N}\right|$, where P is the peak amplitude or charge transfer of IPSCs in the preferred direction, and N is that in the null direction.

## Visual stimulation

A white organic light-emitting display (OLEDXL, eMagin, Bellevue, WA; 800 × 600 pixel resolution, 60 Hz refresh rate) was controlled by an Intel Core Duo computer with a Windows seven operating system and was presented to the retina at a resolution of 1.1 µm/pixel. Moving bar stimuli were generated by MATLAB and the Psychophysics Toolbox (*Brainard, 1997*), and projected through the condenser lens of the two-photon microscope onto the photoreceptor layer. For the flashing spot stimulus in *Figure 3G*, we used 10 repetitions of a 110 µm radius white spot (2 s black, 2 s white, 2 s black) to test On and Off responses. For the simple moving bar stimulus, a positive-contrast bar (110 µm wide, 385 µm long) moved along the long axis in 8 or 12 pseudo-randomly chosen directions at a speed of 440 µm/sec over a 660µm-diameter field on the retina; and three to five trials were recorded for each direction. The percent stimulus contrast was calculated as $(L_{stimulus} - L_{background})/(L_{stimulus} + L_{background})$. Unless otherwise noted (i.e., *Figure 7D*), the intensity of the moving bar was ~$6.3 \times 10^4$ isomerizations (R*)/rod/s, lying in the photopic range, and the background intensity was ~1800 R*/rod/s, lying at the lower end of the photopic range. Individual checker size was 55×55 µm with each checker's intensity equal to either the background (0) or a white value (expressed as a percentage of the moving bar intensity) drawn from a binomial distribution. Checker pattern was updated at 15 Hz.

## Two-photon calcium imaging of GCaMP6 fluorescence in SACs

Genetically encoded calcium indicator CCaMP6m was expressed in sparse populations of On and Off SACs by intravitreal injection of an AAV vector carrying floxed GCaMP6m (University of Pennsylvania Vector Core) into control (*C*) and KO mice (*Gabra2 flox/flox C* and *Slc32a1 flox/flox C*). GCaMP6 fluorescence of isolated retinas in oxygenated Ames at 32–33°C was imaged in a customized two-photon laser scanning fluorescence microscope (Bruker Nano Surfaces Division). GCaMP6 was excited by a Ti:sapphire laser (Coherent, Chameleon Ultra II, Santa Clara, CA) tuned to 920 nm, and

the laser power was adjusted to avoid saturation of the fluorescent signal. Onset of laser scanning induces a transient On response in On SACs that adapts to the baseline in ~3 s. Therefore, to ensure the complete adaptation of this laser-induced response and a stable baseline, visual stimuli were given after 10 s of continuous laser scanning. To separate the visual stimulus from GCaMP6 fluorescence, a band-pass filter (Semrock, Rochester, MA) was placed on the OLED to pass blue light peaked at 470 nm, while two notched filters (Bruker Nano Surfaces Division) were placed before the photomultiplier tubes to block light of the same wavelength. Imaging was performed in the regions of the retina that contain sparsely labeled SAC varicosities so that individual varicosities could be resolved and the orientation of the dendrites relative to the soma could be determined. The objective was a water immersion objective (60x, Olympus LUMPlanFl/IR). Time series of each imaging window (~20×20 μm) were collected at 30–40 Hz.

## Imaging analysis

Analysis was performed using ImageJ and MATLAB. Circular regions of interest (ROIs) corresponding to individual varicosities and a background region with no GCaMP6 expression were manually selected for each imaging window in ImageJ. The fluorescent time course of each ROI was determined by averaging all pixels within the ROI. The fluorescence of the background region was subtracted from the raw fluorescent signals of the ROIs in the same imaging window at each time frame. The resulting fluorescence measurements were then used to calculate the visually evoked responses in varicosities. The relative fluorescence change in each varicosity was taken as $\Delta F = \frac{F - F_0}{F_0}$, where F is the peak amplitude of fluorescence and $F_0$ is the baseline fluorescence level. Mean $\Delta F$ was obtained by averaging 3–5 trials for each direction. To assess direction selectivity of SAC distal dendrites, we used the vector sum or direction selectivity index (DSI), defined as $\left| \frac{\Delta F_{cf} - \Delta F_{cp}}{\Delta F_{cf} + \Delta F_{cp}} \right|$, where $\Delta F_{cf}$ is the relative fluorescence change in centrifugal motion and $\Delta F_{cp}$ is that in centripetal motion. For statistical analysis, we compared variance across varicosities on the dendrites of the same SAC versus variance across varicosities of different SACs. Since the latter was larger, we averaged DSI and vector sum values of all the varicosities belonging to one SAC to get a single data point, and took N in the statistical analysis to be the number of cells, which approximately equals the number of imaging windows.

## Two-photon targeted loose-attached recording of *GFP*-positive neurons for light response

The two-photon targeted recording of light responses was performed as in (*Pei et al., 2015*). Briefly, the retinas were perfused with oxygenated Ames at 32–33°C. Cells were visualized with infrared light (>900 nm) and an IR-sensitive video camera (Watec). *Drd4-GFP*-positive cells were identified using a two-photon microscope (Bruker Nano Surface Division) and a Ti:sapphire laser (Coherent Chameleon Ultra II) tuned to 920 nm. An electrode of 3–5 MΩ was filled with Ames' medium for loose patch recordings of spikes. Then the electrode was carefully removed, and a new electrode filled with cesium-based internal solution and 25 μm Alexa 594 was used to fill the recorded cell. An image stack of the Alexa-488 filled pDSGC was acquired with the two-photon microscope at *z* intervals of 1.5 μm and was resampled three times for each z-plane using a 60x objective (Olympus LUMPlanFl/IR 60x/0.90W). Images were acquired to cover the entire dendritic field of the cell to verify the bis-tratified dendritic arbor and the cofasciculation with *tdTomato*-expressing SACs. Rarely, we encountered *Drd4-GFP* positive neurons that did not show the above two characteristics in both control and knockout groups. These mislabeled cells are not included in the analysis.

Data were analyzed using custom protocols in MATLAB. The number of spikes evoked by the leading (On) and trailing (Off) edges of the bar were counted using MATLAB and averaged across trials in each direction. Spiking DSI is defined as $(N_{pref} - N_{null})/(N_{pref} + N_{null})$ where N is the number of spikes evoked by the leading or trailing edge of the moving bar.

## Statistical analysis

Grouped data with error bars are presented as mean ± SEM and tested for normality. For *Figures 2*, *3*, *5* and *7*, statistical differences were examined using one-way analysis of variance and *post hoc* comparisons using Student's *t*-test with Bonferroni correction. For *Figures 4* and *6*, a two-sample

Kolmogorov–Smirnov test was used and multiple comparisons were corrected by Bonferroni correction.

## Acknowledgements

We thank Chen Zhang for managing a mouse colony and intravitreal injections, Dr. Uwe Rudolph from Harvard Medical School for the generous gift of the *Gabra2*$^{flox/flox}$ mouse line, and the Genetically-Encoded Neuronal Indicator and Effector (GENIE) Project and the Janelia Research Campus of the Howard Hughes Medical Institute GENIE Program and the Janelia Research Campus, specifically Vivek Jayaraman, Ph.D., Douglas S Kim, Ph.D., Loren L Looger, Ph.D., Karel Svoboda, Ph.D. for the AAV-GCaMP6 vectors. This work was supported by NIH R01 EY024016, E. Matilda Ziegler Foundation Grant, Whitehall Grant, Sloan Research Fellowship, Karl Kirchgessner Foundation Grant and Brinson Foundation Award. The authors declare no competing financial interests.

## Additional information

### Funding

| Funder | Author |
| --- | --- |
| National Eye Institute | David Koren<br>Wei Wei |
| Whitehall Foundation | Wei Wei |
| E. Matilda Ziegler Foundation for the Blind | Wei Wei |
| Karl Kirchgessner Foundation | Wei Wei |
| Alfred P. Sloan Foundation | Wei Wei |

The funders provide financial support to this manuscript in study design, data collection and interpretation, and the decision to submit the work for publication.

### Author contributions

QC, WW, Conception and design, Acquisition of data, Analysis and interpretation of data, Drafting or revising the article; ZP, Conducted experiments in Figure 3; DK, Acquisition of data, Analysis and interpretation of data

### Author ORCIDs

Wei Wei, http://orcid.org/0000-0002-7771-5974

### Ethics

Animal experimentation: All procedures to maintain and use mice were in accordance with the University of Chicago Institutional Animal Care and Use Committee (Protocol number ACUP 72247) and in conformance with the NIH Guide for the Care and Use of Laboratory Animals and the Public Health Service Policy.

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
