## [Decision Letter]

Thank you for submitting your article "Stimulus-dependent recruitment of lateral inhibition underlies retinal direction selectivity" for consideration by *eLife*. Your article has been favorably evaluated by Andrew King (Senior Editor) and three reviewers, one of whom is a member of our Board of Reviewing Editors. The following individual involved in review of your submission has agreed to reveal her identity: Yang Dan (Reviewer #3).

The reviewers have discussed the reviews with one another and the Reviewing Editor has drafted this decision to help you prepare a revised submission.

Summary:

This study provides key insights into the mechanisms underlying direction selectivity. The specific question addressed in this study concerns the role of lateral inhibition between starburst amacrine cells (SACs), the inhibitory interneurons that provide a source of directional inhibition onto direction selective ganglion cells. Using precise genetic manipulations to SACs, they provide direct evidence that the circuitries of ON and OFF SACs must indeed be different, as they rely on GABAergic input to very different degrees. The most exciting result of this study, however, is that SAC-SAC interactions render SAC direction selectivity more robust under more natural stimulus conditions, that is in the presence of noise. This is, to our knowledge, the first convincing demonstration of the long standing idea that different dendritic direction selectivity mechanisms supplement each other in order to render motion direction detection robust and as independent as possible from the stimulus.

All reviewers agree that this is a very nice, carefully conducted study with important results. The effects are clear and there are several controls based on paired recordings to assess the physiological impact of the knockouts.

Essential revisions:

1) The finding that lateral inhibition in the ON pathway is required for direction selectivity only with noisy background is intriguing, and the study would be greatly strengthened if the authors can expand on that point more. In particular, the authors are drawing this conclusion on a fairly small parameter set. Although they vary the contrast of the random checkerboards, they use only one bar velocity and one bar contrast for the primary stimulus. This is potentially dangerous since the changing checkerboard intensity is effectively changing the contrast between bar and background. This is particularly important for the recording conditions since at 32 °C (slightly below physiological temperatures) the photoreceptor responses are slower and 15 Hz noise refresh rate for the checkerboard might be equivalent to a gray background.

We recommend that the authors strengthen this particular aspect of the study. Since all the methods are in place, such additional experiments should be doable within a few weeks.

A few suggestions would be:

Alter the contrast and/or velocity of the stimulus and assess the impact of flickering background on tuning in WT and GABA-A α2-ko.

Report whether the noisy stimulus paradigm impacts the tuning in Vgat KO mice. This would greatly strengthen the argument (subsection “The role of lateral inhibition during a moving bar stimulus against a noisy background”) that SAC-SAC inhibition is important for safeguarding direction selectivity in noisy environment.

In Figure 7, control and KO showed no difference at 0% but a large difference at 3% checker intensity, so it seems like a sudden increase. It would be helpful to map that part of the curve a bit more to see the threshold at which inhibition starts playing a role.

2) It would be helpful to add a bit more discussion on why lateral inhibition between starburst cells would be most effective in a noisy background.

3) The reference list reads as if there was no research done on direction selectivity before the year 2000; in fact, the only reference to earlier work is a methods paper on visual stimulus software. This is particularly prominent in their Introduction. The authors should acknowledge earlier work appropriately.

4) One mystery that is not completely explained is why a decrease in lateral inhibition leads to a decrease in SAC responses to centripetal motion, enhancing direction selectivity in the OFF pathway. The reason for this is not at all clear and would be helpful for the authors to postulate the reason for this.

5) The figures are underselling the great quality of the data – the accessibility of the data could be much improved by cleaning up the figures and making them more consistent. Fonts are often too small, so are some of the illustrations; the use of colors is partially inconsistent, and repetitive, unnecessary text clutters the figures (e.g. Figure 6). In addition, in some of the figures the use of abbreviations adds work for non-expert readers. For example, in Figure 2, "CP" and "CF" could have been spelled out the first time they are used since there is space. These terms are explained in the text, but one did have to spend time searching for them.

6) Report light intensities used at the retina level. State explicitly whether the light stimulus conditions were in the photopic range.

7) Report the bath temperature for the whole-cell voltage clamp recordings. Were they also at 32-33 °C?

---

## [Author Response]

Essential revisions:

1) The finding that lateral inhibition in the ON pathway is required for direction selectivity only with noisy background is intriguing, and the study would be greatly strengthened if the authors can expand on that point more. In particular, the authors are drawing this conclusion on a fairly small parameter set. Although they vary the contrast of the random checkerboards, they use only one bar velocity and one bar contrast for the primary stimulus. This is potentially dangerous since the changing checkerboard intensity is effectively changing the contrast between bar and background. This is particularly important for the recording conditions since at 32 °C (slightly below physiological temperatures) the photoreceptor responses are slower and 15 Hz noise refresh rate for the checkerboard might be equivalent to a gray background.

We recommend that the authors strengthen this particular aspect of the study. Since all the methods are in place, such additional experiments should be doable within a few weeks.

A few suggestions would be:

Alter the contrast and/or velocity of the stimulus and assess the impact of flickering background on tuning in WT and GABA-A α2-ko.

Report whether the noisy stimulus paradigm impacts the tuning in Vgat KO mice. This would greatly strengthen the argument (subsection “The role of lateral inhibition during a moving bar stimulus against a noisy background”) that SAC-SAC inhibition is important for safeguarding direction selectivity in noisy environment.

In Figure 7, control and KO showed no difference at 0% but a large difference at 3% checker intensity, so it seems like a sudden increase. It would be helpful to map that part of the curve a bit more to see the threshold at which inhibition starts playing a role.

We appreciate that the reviewers raised this important question: Is adding the flickering checkerboard background equivalent to simply changing the contrast of the moving bar stimulus? If it is, we should see reduced direction selectivity in the On pathway in *Gabra2* KO mice during a moving bar stimulus with contrast and background luminance levels that are similar to the equivalent flickering checkerboard stimulus. Thus, we performed this experiment and found that direction selectivity in the On pathway in *Gabra2* KO mice is not affected at all contrast and background luminance levels we tested. Therefore, lateral inhibition in the On pathway is functionally recruited by the noisy background but not by the specific contrast of the primary stimulus nor the background luminance levels. This data has now been included in Figure 7, and in the text (subsection “Lateral inhibition in the On pathway is required for direction selectivity on a noisy background”, last paragraph).

We agree that impaired tuning of DSGCs during a noisy stimulus in the absence of SAC-SAC inhibition would directly support a role for reciprocal SAC inhibition. We have previously recorded from DSGCs from Vgat KO mice and have found that the tuning of DSGCs is significantly impaired (Pei et al., 2015). However, in these KO mice, the SAC-DSGC inhibition is also disrupted. Therefore, we cannot assign the reduced direction selectivity to SAC-SAC inhibition. Since we currently cannot provide direct support for a role of SAC-SAC inhibition in a noisy background, we have modified the Discussion section accordingly (subsection “The role of lateral inhibition during a moving bar stimulus against a noisy background”).

We are also technically limited in that our visual stimulation setup cannot reliably produce checker intensities lower than 3% of the moving bar brightness used in this study. We will work on overcoming these technical difficulties for future studies.

2) It would be helpful to add a bit more discussion on why lateral inhibition between starburst cells would be most effective in a noisy background.

We have included more discussion in the subsection “The role of lateral inhibition during a moving bar stimulus against a noisy background”.

3) The reference list reads as if there was no research done on direction selectivity before the year 2000; in fact, the only reference to earlier work is a methods paper on visual stimulus software. This is particularly prominent in their Introduction. The authors should acknowledge earlier work appropriately.

We apologize for the missing references and have added them to the text (Introduction, first paragraph and subsection “The role of lateral inhibition during a moving bar stimulus against a noisy background”).

4) One mystery that is not completely explained is why a decrease in lateral inhibition leads to a decrease in SAC responses to centripetal motion, enhancing direction selectivity in the OFF pathway. The reason for this is not at all clear and would be helpful for the authors to postulate the reason for this.

We have discussed our current hypothesis on the improved direction selectivity of the Off pathway in Vgat KO mice in the Discussion section (subsection “The Off pathway”, second paragraph).

*5) The figures are underselling the great quality of the data – the accessibility of the data could be much improved by cleaning up the figures and making them more consistent. Fonts are often too small, so are some of the illustrations; the use of colors is partially inconsistent, and repetitive, unnecessary text clutters the figures (e.g. Figure 6). In addition, in some of the figures the use of abbreviations adds work for non-expert readers. For example, in Figure 2, "CP" and "CF" could have been spelled out the first time they are used since there is space. These terms are explained in the text, but one did have to spend time searching for them.*

We have modified the figures in the following ways:

A) Removed redundant text and scale bars in Figure 1, Figure 2, Figure 3, Figure 4 and Figure 6.

B) Increased the smaller fonts and illustrations in Figure 2 and Figure 3.

C) Changed the colors of the bar graphs to be more consistent: empty columns are control groups, and filled columns are knockout groups.

D) Changed "CP" and "CF" to "Centripetal" and "Centrifugal" in Figure 2 and Figure 4.

6) Report light intensities used at the retina level. State explicitly whether the light stimulus conditions were in the photopic range.

The light intensity for the moving bar is ~ 6.3 x 10^4^ R*/rod/s. It is in the photopic range. We have noted the light intensities of our visual stimuli in the "Experimental procedures" and in Figure 7.

7) Report the bath temperature for the whole-cell voltage clamp recordings. Were they also at 32-33 °C?

All light-evoked whole-cell and loose-patch recordings were performed at bath temperatures of 32-33 °C. We have included this information in the text (subsection “Whole-cell voltage-clamp recording”, first paragraph).